# Haplotype of Wild Korean Boars Infected by Classical Swine Fever Virus Subgenotype 2.1d

**DOI:** 10.3390/ani12192670

**Published:** 2022-10-04

**Authors:** SeEun Choe, Ki-Sun Kim, Gyu-Nam Park, Sok Song, Jihye Shin, Bang-Hun Hyun, Dong-Jun An

**Affiliations:** Virus Disease Division, Animal and Plant Quarantine Agency, Gimcheon 39660, Gyeongsangbuk-do, Korea

**Keywords:** CSFV, phylogenetic tree, haplotype, mtDNA, wild boar

## Abstract

**Simple Summary:**

Classical swine fever is a highly contagious disease that infects both domestic pigs and wild boars. Classical swine fever virus (CSFV) has not been detected in domestic pigs in South Korea since 2016, but has been increasing in wild boars since 2017. Two cases of CSFV subgenotype 2.1d were detected in wild Korean boars in 2011, but then no cases were detected until 2016; however, 16 cases of CSFV were detected between 2017 and 2019. In this study, we report seven CSFV-positive samples obtained from wild boars in 2020. In addition, although 13 mtDNA haplotypes were detected in wild boars in South Korea, all 25 cases of CSFV that occurred in wild boars between 2011 and 2020 were detected in animals with haplotype 01.

**Abstract:**

Classical swine fever virus (CSFV) is one of the major pathogens that causes severe economic damage to the swine industry. Circulation of CSFV in wild boars carries the potential risk of reintroducing the virus into CSFV-free pig farms. This study carried out a genetic analysis of CSFV isolates from wild boars and analyzed the mtDNA haplotypes of the wild boars. Blood samples (n = 2140) from wild Korean boars captured in 2020 were subjected to qRT-PCR to detect CSFV, which was classified as subgenotype 2.1d based on phylogenetic analysis. CSFV had been detected in wild boars only in northern regions (Gangwon and Gyeonggi) of South Korea between 2011 and 2019. However, CSFV was identified in wild boars in the more southern regions (Chungbuk and Gyeongbuk) in 2020. Based on mitochondrial DNA analysis, all wild boars with CSFV were haplotype 01 (H01). Thus, we presume that the H01 haplotype is more susceptible to CSFV. In the future, infection of wild boars by CSFV is expected to occur intermittently every year, and we predict that most wild boars infected with CSFV will be haplotype H01.

## 1. Introduction

Classical swine fever virus (CSFV; genus *Pestivirus*, family *Flaviviridae*) is a single-stranded, positive-sense RNA virus with a genome of approximately 12,300 nucleotides. Classical swine fever (CSF) is a highly contagious and fatal multisystemic hemorrhagic disease of swine (*Sus scrofa*) that affects both breeding pigs and wild boars [1]. CSFV detection in wild boars has been reported not only in South Korea [2,3,4] but also in European countries [5,6,7] and Japan [8]. Wild boars transmit CSF to domestic pigs in both Japan and South Korea [3,8]. Two CSFV isolates were first recovered from wild Korean boars in 2011. Since then, three, two, and eleven CSFV isolates were reported in wild boars captured in 2017, 2018, and 2019, respectively [3,4]. All 18 CSF viruses isolated from 2011 to 2019 are classified as subgenotype 2.1d. Among these, the YC11WB strain isolated in 2011 was confirmed to be a virulent virus by the challenge test [3,4]. Between 2017 and 2019, CSFV spread from the northwest (Gyeonggi) to the northeast (Gangwon) of South Korea, which are adjacent to each other [4]. However, there are few data about current CSFV occurrence patterns and strain characteristics, and no studies have attempted to predict future CSF outbreaks in South Korea.

A previous study shows that wild Korean boars belong to the Asian wild boar cluster [9]. Mitochondrial DNA (mtDNA) is a suitable genetic marker for studying genetic variations within a species because it is inherited from the mother without recombination [9]. To date, few studies have examined the links between CSFV infection and wild boar haplotype.

Here, we investigated the CSFV genetic type and recent status of CSF occurrence in wild boars, as well as attempting to predict future CSF epidemics. Additionally, we examined the mtDNA haplotype of captured wild boars to investigate which genetic type is susceptible to CSFV infection.

## 2. Materials and Methods

### 2.1. RT-PCR Amplification of CSFV

From 2010, wild boars were hunted in cooperation with the Korean Pork Producers Association and the Korean government to satisfy the OIE requirements for surveillance of wild boars and feral pigs in CSF-free countries. Blood samples were collected from 2140 wild boars hunted in nine provinces (Gangwon, Gyeonggi, Gyeongnam, Gyeongbuk, Jeonnam, Jeonbuk, Chungnam, Chungbuk, and Jeju) in South Korea in 2020 (Table 1) and subjected to quantitative real-time (qRT)-PCR analyses to detect CSFV. Total RNA was extracted using a microcolumn-based RNeasy mini kit (Qiagen, Hilden, Germany) and subjected to VDx CSFV qRT-PCR (Median Diagnostic Co., Cat No. NS-CSF-31, Chuncheon, South Korea), which employs TaqMan probes to detect the CSFV 5’ UTR with high specificity. This assay does not detect bovine viral diarrhea virus or border disease virus, which also belong to the genus *Pestivirus*. The PCR assays for CSFV were performed as described previously [10]. The complete envelope glycoprotein E2 gene sequences were amplified by reverse transcription PCR (RT-PCR) as described previously [3,11].

### 2.2. Phylogenetic Tree and Bayesian Skyline Analysis of CSFV

The complete E2 gene sequences of reference CSFVs were obtained from the National Center for Biotechnology Information (NCBI) GenBank database and aligned using the CLUSTAL X alignment program. A BEAST input file was then generated using BEAUti within the BEAST package v1.8.1 [12]. Rates of nucleotide substitution per site and per year, and time to the most recent common ancestor (tMRCA), were estimated using a Bayesian Markov Chain Monte Carlo (MCMC) approach. The resulting convergence was analyzed using Tracer 1.5 [13]. Trees were summarized as a maximum clade credibility (MCC) tree using TreeAnnotator 1.7.4 [14] and visualized using Figtree 1.4 [15]. The exponential clock and expansion growth population model in the BEAST program was used to obtain the best-fit evolutionary model, and the MCC tree was visualized using Figtree 1.4 [15]. Estimated divergence time and 95% highest posterior density (HPD) intervals, which summarize statistical uncertainties, were indicated at each tree node. Changes in effective population size were plotted using Bayesian skyline plot (BSP) analyses [16].

### 2.3. PCR of the D-Loop Region of Wild Boar mtDNA

A total of 347 wild boar blood samples (322 samples collected during nationwide monitoring in 2020 and 25 CSF-positive wild boar samples obtained between 2011 and 2020) were subjected to PCR amplification of the D-loop region of mtDNA. To obtain uniform nationwide data, a total of 322 samples were selected so that at least five wild boars could be included, centered on cities located within nine provinces (Gangwon, Gyeonggi, Gyeongnam, Gyeongbuk, Jeonnam, Jeonbuk, Chungnam, Chungbuk, and Jeju). The 322 blood samples used for mtDNA PCR originated from 2140 wild boars captured in different regions of South Korea during a 2020 CSFV surveillance campaign. In addition, we investigated the haplotype of 25 CSF-positive wild boars that were confirmed to be CSF-positive between 2011 and 2020. The mtDNA D-loop region PCR, which targets the highly conserved tRNA-Pro and tRNA-Phe sequences, was performed as previously described [9]. Briefly, total DNA was extracted from the blood of wild boars using the DNeasy blood and tissue kit (Qiagen, Cat No. 69504, US). PCR of the mtDNA D-loop was performed using the AccuPower ProFi taq PCR PreMix (Bioneer Inc., Cat No. K-2632, South Korea) and primers pDF (5′-AGCACCCAAAGCTGAAATTC-3′) and pDR (5′-AGCTGTGAGGCTCATCTAGG-3′). The specific primers amplified a mtDNA fragment of 1274 bp (GenBank accession number NC_000845; nucleotide positions 16576–1236). The PCR conditions were as follows: initial denaturation at 95 °C for 9 min, 45 cycles of denaturation at 94 °C for 30 s, annealing at 55 °C for 30 s, extension at 72 °C for 30 s, and a final elongation step at 72 °C for 5 min. PCR reactions were performed using the AccuPower ProFi Taq PCR PreMix kit (Bioneer Inc., Cat No. K-2632, Daejeon, South Korea).

### 2.4. Phylogenetic and Network Analysis of Haplotypes

For phylogenetic analysis, 428 mtDNA D-loop region sequences, which included 81 wild Korean boar reference mtDNA sequences from GenBank (KY911599, KY911609-KY911636, KY911642-KY911653, KY911665, KY911671-KY911691, KY911693-KY911699, and JN251947-JN251957) and 347 wild boar samples from this study, were aligned using CLUSTAL X (ver. 2.1.) [17]. Phylogenetic trees based on partial mtDNA D-loop gene sequences (683 bp) were reconstructed using the maximum likelihood (ML) method in MEGA 7.0 software [18]. The Tamura–Nei model (gamma-distributed with invariant site: G + I) was used, with 1000 replications for bootstrap analysis. The number of haplotypes (h), haplotype diversity (Hd), and nucleotide diversity (*π*) were computed using DnaSP 6.12.03 [19]. The Tajima D test [20,21] and Fu’s Fs [22] were calculated from 1000 simulated samples to demonstrate selective neutrality or population demographic expansion. Harpending’s raggedness index (*r*) [23] was calculated based on the maximum number of mutational differences and the frequencies of the allelic classes. Relationships among haplotypes of wild Korean boars were estimated using a parsimonious median-joining method [24] and visualized using Network 4.5.1.0 and Network Publisher 1.1.0.6 (http://www.fluxus-engineering.com accessed on 30 December 2021).

## 3. Results

### 3.1. The E2 Gene of CSFV from Wild Korean Boars

RT-PCR and qRT-PCR of the E2 gene identified seven CSFV strains from 2140 wild Korean boars captured during the 2020 CSFV surveillance campaign. The prevalence of CSF in wild boars in 2020 was 0.33% (7/2140) (Table 1). The seven CSFVs were detected in Gangwon (HC20WB01, -02, -03, DH20WB01, and WJ20WB01), Chungbuk (CJ20WB01), and Gyeongbuk (MK20WB01 (Table 1 and Table 2, Figure 1). The E2 gene nucleotide (nt) and amino acid (aa) sequence homology of the seven CSFVs was 98.9–99.7% and 99.2–100%, respectively. A comparison of the seven CSFVs identified in 2020 with the two CSFVs (YC11WB, PC11WB) first isolated in 2011 revealed that the seven recent isolates were similar to YC11WB with a homology percentage of 95.95% ± 0.15% and 98.4% ± 0.3% at the nt and aa level, respectively, and to PC11WB with a homology of 97.2% ± 0.1% and 99.2% ± 0.3% at the nt and aa level, respectively (Table 2).

### 3.2. Phylogenetic Tree of CSFV

The Bayesian tree revealed that the partial E2 (1119 bp) sequences of 70 CSFVs, which included 25 strains detected in wild Korean boars between 2011 and 2020, clustered into three genotypes (G1, G2, and G3). These genotypes were classified into seven subgenotypes (G1.1 and G1.4; G2.1, G2.2, and G2.3; and G3.2 and G3.4) (Figure 1). Subgenotype G2.1 was divided further into G2.1a, G2.1b, G2.1c, and G2.1d. The 25 CSFV strains detected in wild Korean boars between 2011 and 2020 belonged only to the G2.1d subgenotype (Figure 1). CSFV (SW2003) isolated from South Korean domestic pigs in 2003 was also the G2.1b subgenotype, but YC16CS isolated in 2016 was reclassified as the G2.1d subgenotype (Figure 1). The geometric mean of tMRCA was 2,489,772, and the 95% HPD interval was 1,987,174 (upper) and 3,067,167 (lower). The clock rate geometric mean was 6.8853 × 10^−4^, and the 95% HPD interval was 5.3886 × 10^−4^ (upper) and 8.4995 × 10^−4^ (lower). The effective sample size (ESS) was 1217 (Figure 1).

### 3.3. Prediction of CSF Occurrence in Wild Boars

The yearly breakdown of the 25 CSFV isolates collected from wild boars between 2011 and 2020 was as follows: two in 2011, three in 2017, two in 2018, eleven in 2019, and seven in 2020 (Table 2). The geographic distribution of the 25 CSFV isolates was as follows: sixteen in Gangwon (GG), seven in Gyeonggi (GG), one in Chungbuk (CB), and one in Gyeongbuk (GB) (data not shown). A Bayesian skyline graph of the 25 wild Korean boar CSFV strains isolated from 2011 to 2020, which included 18 reference strains (2011–2019) from GenBank, showed a sharp increase beginning in 2018 (Figure 2). The geometric mean of tMRCA was 31.7483, and the 95% HPD interval was 17.4029 (upper) and 53.9031 (lower). The mean level (95% HPD upper) of the Bayesian skyline was 15.98–25.13 (38.9–81.34) from 2002 to July 2018, but increased to 30.59–47.90 (98.24–218.23) from September 2018 to December 2020 (Figure 2). The geometric mean clock rate was 8.0416 × 10^−4^, and the 95% HPD interval was 3.5242 × 10^−4^ (upper) and 1.4673 × 10^−3^ (lower). The ESS was 1480. BSP analysis predicted that occurrence of CSF in wild boars in South Korea will increase in the future.

### 3.4. Distribution and Nucleotide Diversity of mtDNA Haplotypes among Wild Korean Boars

The mtDNA sequences from 428 wild Korean boars, which included 81 reference sequences, were classified into 13 haplotypes (H01–H13) as follows: 136 haplotype 1 (H01), six H02, 70 H03, two H04, five H05, two H06, two H07, 34 H08, 78 H09, 25 H10, two H11, 65 H12, and one H13 (Table 1 and Figure 3). Haplotype distribution varied by region, but those of the Jeju (JJ) region were classified as a single type (H12) (Figure 3). The predominant haplotypes in each region were as follows: H08 (16/38) in GG, H01 (51/82) in GW, H01 (16/33) in Chungnam (CN), H03 (7/20) in CB, H01 (42/61) in GB, H09 (25/57) in Gyeongnam (GN), H03 (8/23) in Jeonbuk (JB), H09 (34/47) in Jeonnam (JN), and H12 (65/65) in JJ (Figure 3). In addition, the H01 type identified in wild boars from the Pyeonganbukdo (PB) and Hwanghaebukdo (HB) regions in North Korea was used as a reference (Figure 3). The mtDNA haplotypes of the 25 CSFV-positive wild Korean boars from 2011 to 2020 were all H01. Regarding wild Korean boar haplotype genetic variability, haplotype diversity (Hd), which was estimated by analyzing the 683 bp fragment control region of mtDNA, ranged from 0.030 in JJ to 0.759 in JB (Table 3). The nucleotide diversity (π) of wild Korean boar mtDNA ranged from 0.00054% in JJ to 0.00932% in GG (Table 3). The value of Tajima’s D ranged from 0.00713 in JJ to 2.91511 in CB, and Fu’s Fs ranged from 1.422 in JJ to 8.371 in GG (Table 3). The r statistic ranged from 0.1404 in GN to 0.9421 in JJ (Table 3).

### 3.5. Parsimonious Median-Joining Network and Phylogenetic Analyses of mtDNA from Wild Korean Boars

The parsimonious median-joining network and phylogenetic tree analysis using the ML method revealed that the 13 haplotypes (H01—H13) grouped into three clusters (Figure 4A,B). The first (H01, H02, H07, H09, and H13) and second (H03, H04, H06, and H10) clusters were distributed nationwide, except for JJ (Figure 4B). The third cluster (H05, H08, H11, and H12) was found in all regions except for JN, JB, and CB (Figure 4B). A comparison of nucleotide substitutions in mtDNA among the three clusters revealed that the number of predicted differential marker nucleotide sequences was three in cluster I (G-427, T-446, and G-676), one in cluster II (C-198), and two in cluster III (C-165 and C-266) (data not shown).

## 4. Discussion

Wild boars play an important epidemiological role in CSFV transmission in both wild boars and domestic pigs. Data from the World Organization for Animal Health released in 2022 show that there are now approximately 38 CSFV-free countries, including the United States, Canada, Brazil, Chile, South Africa, and EU countries (www.oie.int accessed on 2 August 2022). There has, however, been a disturbing trend of recurrence in some countries that had declared CSF to be eliminated (i.e., France, the Netherlands, Germany, and Belgium); this recurrence is due to wild boars [25,26]. In Europe, fencing, hunting, trapping, and oral vaccination are implemented as measures to prevent the spread of CSFV by wild boars [25]. There are reports that CSFV can spread directly or indirectly from wild boars to livestock [25]. In Germany, 60% of the 92 cases were caused by direct or indirect contact with a wild boar [27]. In 2018, CSF reemerged in the Gifu Prefecture of central Japan, causing an ongoing outbreak in wild boars and domestic pigs [28,29]. Thereafter, CSFV spread rapidly to neighboring prefectures, affecting both domestic pigs and wild boars [30]. Since the start of the wild boar policy project in South Korea, which aims to eliminate CSF, the disease has been detected in 18 wild boars captured between 2011 and 2019 [3,4], and seven wild boars tested positive for CSFV in 2020. South Korea is divided into nine provinces, and between 2011 and 2019, CSF occurred only in wild boars living in the northern provinces of Gyeonggi (GG) and Gangwon (GW). However, the 2020 survey detected one wild boar in each of the Chungbuk (CB) and Gyeongbuk (GB) provinces, which are located below GW province. The number of detected cases, and the transmission rate, of CSFV in wild Korean boars are lower and slower, respectively, than those in wild Japanese boars. However, the reasons underlying the occurrence of CSF in wild boars in regions in which CSF does not occur can be diverse, and various epidemiological investigations should be conducted in the future. The predicted cause of transmission is the large-scale wild boar hunting that is being carried out to eradicate African swine fever virus (ASFV) [31]. It is speculated that causes include wild boars themselves, hunters, hunting vehicles, hunting dogs, fomites, and wild animals. Therefore, it seems likely that CSFV will be transmitted to wild boars in the southern region in the future.

In Japan, virological analysis demonstrated that CSFVs isolated during the CSF outbreak belong to an emerging clade within the 2.1 subgenotype, which is most closely related to CSFVs detected in China [32,33]. The current CSF outbreaks in Japan are driven by circulation of a CSFV strain with moderate pathogenicity. The sequence of this Japanese CSFV strain is almost identical to that of two CSFVs recently isolated in China and Mongolia, which further complicates our understanding of the patterns and causes of CSF outbreaks [8,28]. All wild Korean boar CSFV genotypes are G2.1d, which is similar to the genotype of the YC16CS strain detected in domestic pigs in 2016. This finding confirms transmission and circulation of CSFV between wild boars and domestic pigs [4]. CSF has not occurred in South Korean domestic pigs since 2016 (one case in 2013 and two cases in 2016). The CSFV case in 2013 belonged to 2.1b, whereas that in 2016 belonged to 2.1d [3,4]. Interestingly, the YC16CS strain in 2016 was the first suspected case of infection between wild boars and domestic pigs (because GG province is where CSF was detected in wild boars in 2011) [4]. The wild boar YC11WB strain (CSFV subgenotype 2.1d) isolated in 2011 is highly pathogenic in domestic pigs [34]. However, the recently isolated CSFV subgenotype 2.1d from Japan showed moderate pathogenicity [32,33]. We investigated the strains prevalent in wild Korean boars (isolated between 2017 and 2020) to identify genetic differences (about 4% nt and 2% aa in the E2 gene), even if it belongs to the same subgenotype 2.1d as the YC11WB strain. Future experiments are planned to confirm the pathogenicity in pigs. Based on the results of the CSFV genetic investigations so far, we expect it to have moderate pathogenicity.

Phylogenetic analysis of mitochondrial gene variations in wild Japanese boars provides evidence of distinct lineages between those that inhabit the mainland and those that inhabit the islands [35,36,37]. Other studies also analyzed the geographic population structure and sequence divergence of mtDNA from wild boars and domestic pigs in Japan [38] and Thailand [39]. Three of the eight mtDNA haplotypes detected in Shikoku, Japan, are of a wild Japanese boar and European domestic pig lineage [35]. However, no antibodies against CSFV or Aujeszky’s disease virus were detected in 113 wild boars on Shikoku. This suggests that they were not exposed to those pathogens [35]. Bayesian phylogenetic analysis of the partial mtDNA D-loop region sequences of 56 wild Korean boars using MCMC methods identified 25 haplotypes, which were classified into four distinct subgroups (K1 to K4) [9]. Interestingly, the 25 wild Korean boars infected with CSF were all H01. Although this haplotype accounted for 31.5% of the wild Korean boar population, it was noteworthy that CSFV was not detected in any other haplotypes. This result suggests that H01 may be more susceptible to CSFV infection than other haplotypes. Thus, the H01 haplotype can be a potential risk factor that accelerates CSFV transmission due to its high prevalence in wild Korean boars. To clarify this assumption, more studies are needed in the future. In particular, it should be noted that the distribution of H01 is very low in the southern regions of South Korea, such as Gyeongnam, Jeonbuk, and Jeonnam provinces. It will also be important to identify and analyze CSF occurrence patterns and the haplotypes of wild boars in these regions.

## 5. Conclusions

Wild boars in South Korea will continue to migrate to the southern regions because of large-scale hunting, which is being conducted to eradicate ASFV from the northern regions. Furthermore, the occurrence of CSFV subgenotype 2.1d is expected to spread nationwide. The 25 wild Korean boars infected with CSF were all H01. Although this haplotype accounted for 31.5% of the wild Korean boar population, it is noteworthy that CSFV was not detected in any other haplotypes. This result suggests that CSFV may be more susceptible to H01 than other haplotypes.

## Figures and Tables

**Figure 1 animals-12-02670-f001:**
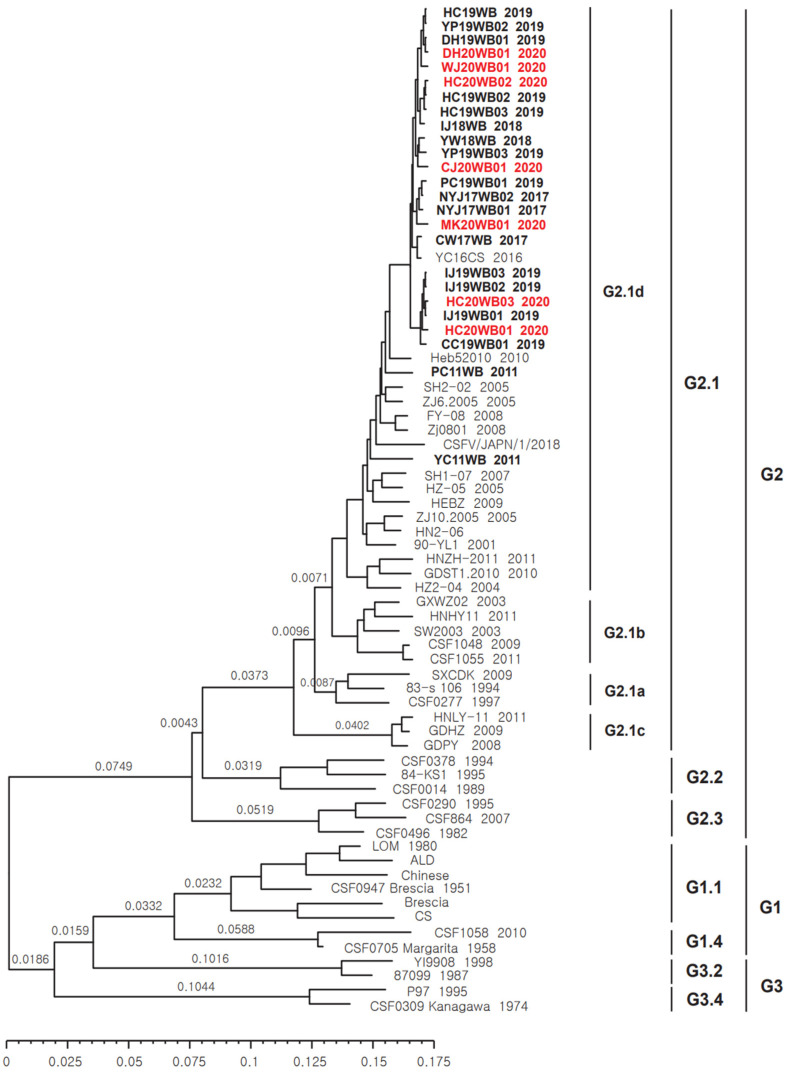
Phylogenetic tree of CSFV partial E2 gene sequences. MCC tree analysis of 70 CSFV strains, including 25 CSFVs isolated from wild Korean boars, was conducted using the BEAST program. CSFVs from wild Korean boars are marked in black bold font (2011–2019) and red bold font (2020). Bar = branch age.

**Figure 2 animals-12-02670-f002:**
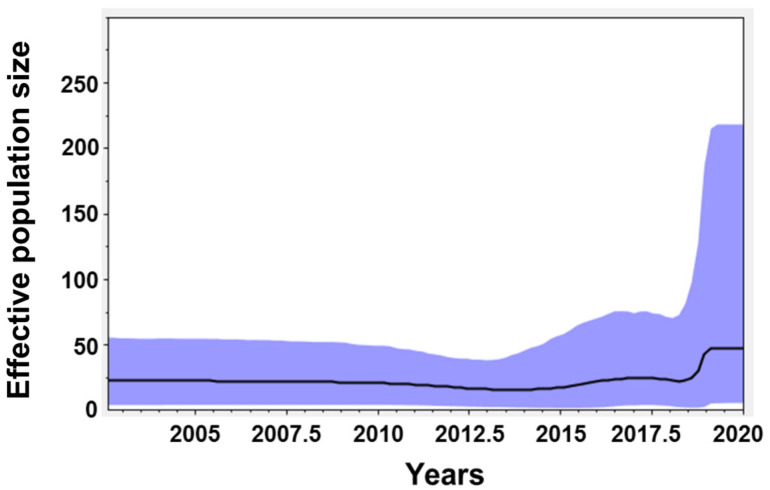
Skyline population analysis of wild Korean boars infected with CSFV. CSFV skyline population analysis of wild Korean boars from 2002 to 2020. The black line denotes the mean level for the effective population size, and the shaded blue region represents the high and low levels for the effective population size.

**Figure 3 animals-12-02670-f003:**
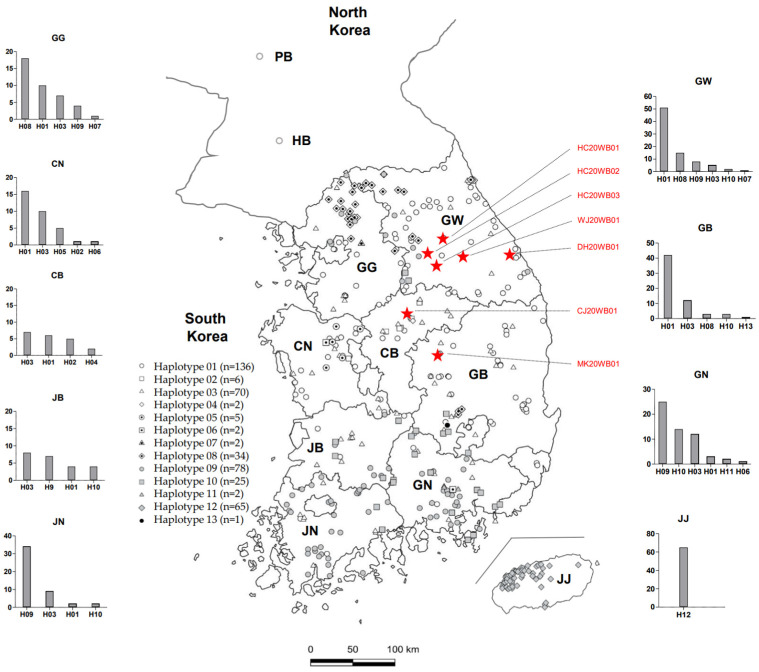
Haplotype distribution of mtDNA sequences from wild boars captured in North Korea and South Korea. Symbols denoting the 13 haplotypes in 428 wild boars are shown on the left side of the map. The kind of haplotypes in each region was graphed on the left and right sides of the map. PB: Pyeonganbukdo; HB: Hwanghaebukdo; GW: Gangwon; GG: Gyeonggi; GN: Gyeongnam; GB, Gyeongbuk; JN: Jeonnam; JB: Jeonbuk; CN: Chungnam; CB: Chungbuk; JJ: Jeju. The red star indicates the location at which wild boars infected with CSFV in 2020 were captured. The graphs for each region indicate the number of haplotypes detected within the nine provinces.

**Figure 4 animals-12-02670-f004:**
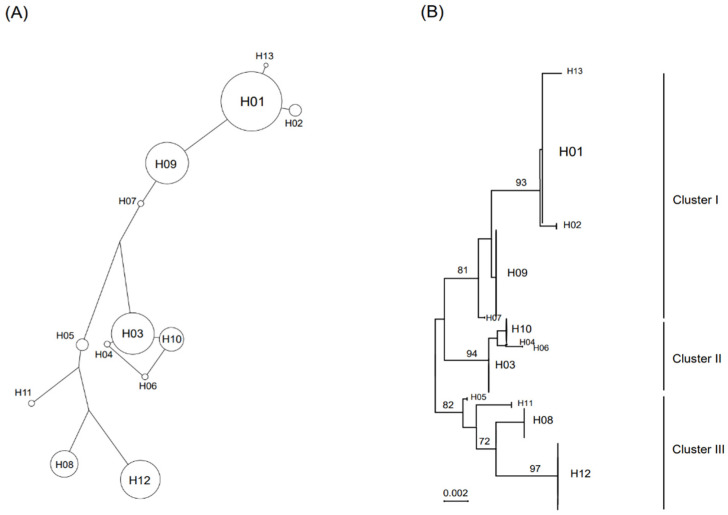
Haplotype network and phylogenetic tree of mtDNA sequences from wild boars captured in South Korea. Parsimonious median-joining networks depicting the relationships among wild Korean boars (**A**). A network illustrating the relationships between wild Korean boar haplotypes. The phylogenetic tree was reconstructed using partial mtDNA D-loop gene sequences from 428 wild Korean boars (**B**). Thirteen haplotypes were identified using the maximum likelihood (ML) method, with 1000 replications used for bootstrap analysis (MEGA 7.0 software). All haplotypes grouped into three clusters (I, II, and III).

**Table 1 animals-12-02670-t001:** Detection of CSFV antigens and haplotype analysis of wild boars.

	Total	North Korea		South Korea								
		PB ^a^	HB	GG	GW	CN	CB	GB	GN	JB	JN	JJ
CSFV	7/2140 ^b^	-	-	0/141	5/201	0/236	1/267	1/486	0/363	0/320	0/82	0/44
Haplotype	12/347 ^c^	-	-	5/30	6/71	5/32	4/20	3/54	6/56	4/23	4/35	1/26
	7/81 ^d^	1/1	1/1	4/8	3/11	1/1	-	3/7	1/1	-	1/12	1/39
	13/428 ^e^	1/1	1/1	5/38	6/82	5/33	4/20	5/61	6/57	4/23	4/47	1/65

^a^ PB, Pyeonganbukdo; HB, Hwanghaebukdo; GG, Gyeonggi; GW, Gangwon; CN, Chungnam; CB, Chungbuk; GB, Gyeongbuk; GN, Gyeongnam; JB, Jeonbuk; JN, Jeonnam; JJ, Jeju. ^b^ Number of CSFV-positive/number of wild boar samples. ^c^ Number of haplotypes/number of wild boar samples examined in this study. ^d^ Number of haplotypes/number of reference wild boar samples examined in the previous study. ^e^ Number of haplotypes/total number of wild boar samples in the previous study and in this study.

**Table 2 animals-12-02670-t002:** Comparison of E2 gene nucleotide and amino acid sequence homologies between wild Korean boar CSFVs isolated from 2017–2020 and two CSFV strains identified in 2011.

2011 Strain	Isolation Year (Number of Strains)			
	2017 (3) ^a^	2018 (2) ^b^	2019 (11) ^c^	2020 (7) ^d^
YC11WB	95.95 ± 0.35 ^e^ 98.0 ± 0.4 ^f^	95.6 ± 0.3 97.85 ± 0.55	95.9 ± 0.2 98.15 ± 0.55	95.95 ± 0.15 98.4 ± 0.3
PC11WB	97.35 ± 0.25 98.8 ± 0.4	96.75 ± 0.25 98.4 ± 0.3	97.15 ± 0.15 98.95 ± 0.55	97.2 ± 0.1 99.2 ± 0.3

^a^ CW17WB (accession number MH548928), NYJ17WB01 (MT027041), NYJ17WB02 (MT027042). ^b^ IJ18WB (MT027043), YW18WB (MT027044). ^c^ HC19WB (MT027045), YP19WB02 (MT027054), YP19WB03 (MT027055), PC19WB01 (MT027051), HC19WB02 (MT02752), HC19WB03 (MT027053), IJ19WB01 (MT027047), IJ19WB02 (MT027048), IJ19WB03 (MT027049), DH19WB01 (MT027050), CC19WB01 (MT027046). ^d^ CJ20WB01, DH20WB01, HC20WB01, HC20WB02, HC20WB03, MK20WB01, WJ20WB01. ^e^ Nucleotide sequence homology (%). ^f^ Amino acid sequence homology (%).

**Table 3 animals-12-02670-t003:** Genetic variability and demographic analysis of mitochondrial DNA from wild boars captured in different geographical regions of North and South Korea.

Country	Location	No. of Wild Boars	mtDNA Control Region ^a^				
			Hd	Π (%)	Tajima’s D	Fu’s Fs	r
Korea ^b^	11 ^c^	428 ^d^	0.807	0.00925	2.63354	8.450	0.0809
South Korea	GG ^e^	38	0.727	0.00932	2.89214	8.371	0.1799
	GW	82	0.573	0.00673	1.48833	7.044	0.3038
	CN	33	0.669	0.00750	1.93526	6.018	0.2823
	CB	20	0.753	0.00758	2.91511	5.791	0.1907
	GB	61	0.490	0.00620	0.70848	6.840	0.3498
	GN	57	0.711	0.00605	0.80308	4.975	0.1404
	JB	23	0.759	0.00635	2.01929	5.320	0.1511
	JN	47	0.446	0.00371	0.34085	4.296	0.4301
	JJ	65	0.030	0.00054	0.00713	1.422	0.9421

^a^ mtDNA control region; mitochondrial DNA control region. ^b^ Both South Korea and North Korea. ^c^ Two regions (Pyeonganbukdo and Hwanghaebukdo) of North Korea and nine regions of South Korea. ^d^ Number of wild boars captured in South Korea (n = 426) and North Korea (n = 2). ^e^ GG, Gyeonggi; GW, Gangwon; CN, Chungnam; CB, Chungbuk; GB, Gyeongbuk; GN, Gyeongnam; JB, Jeonbuk; JN, Jeonnam; JJ, Jeju. Hd, haplotype diversity; Π, nucleotide diversity; r, raggedness statistic.

## Data Availability

The partial D-loop region (683 bp) sequences of 347 mtDNA (accession numbers MZ265820–MZ266166) and the partial E2 gene (1119 bp) sequences of seven CSFV strains (accession numbers MZ332510–MZ332516) detected in wild Korean boars have been deposited in GenBank.

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
