# Peer review of "Haplotype of Wild Korean Boars Infected by Classical Swine Fever Virus Subgenotype 2.1d"

_animals, 2022, doi:10.3390/ani12192670_

Round 1

Reviewer 1 Report

The Title is vague and does not reflect the content of the article.

Authors must establish the goal of the manuscript. In the text, the data presented are not sufficient to support the epidemiological considerations, which presuppose a different approach. The result of laboratory examinations is not sufficient for the reported epidemiological considerations. The same applies for the study on the susceptibility of the specific wild boar genotype to CSF, these evaluations require a different experimental design.

In the title is mentioned (as in the text) that wild boars are migratory animals, which are not. There are several EFSA opinions on the subject. Please, check.

The possible territorial spread of CSFV can be the result of movements of infected pigs / wild boars, products of contaminated pigs or wild boars, people, fomites, vehicles, feed, not only through the movements of wild boars. Given the situation and the spread of CSF in South Korea (the disease is present in domestic pigs and wild boars), the other ways of spreading should also be considered.

Are there structured surveillance plans available whose results allow you to say that the areas where you believe the positivity is shifting were previously disease free?

It should be explained in what context the samples you refer to are collected

References are old and should be updated. As an example, in the European Union countries CSF was eradicated.  Updated data can be found on the WOAH/OIE website.

Line 45: please, support this statement by a reference

Paragraph 3.3: the content does not reflect the name of the title of the paragraph

Line 265-266: did you establish the epidemiological meaning of these positive in relation to the existing wild boar population?

Lines 280-281: This sentence could indicate the presence of the disease in an endemic form in the country.

Lines 284-286: this sentence should be supported by data or specific reference

Lines 299-303: the same as above

Author Response

Review 1

Comment 1: The Title is vague and does not reflect the content of the article.

Answer: Thank you. We have changed the title: original title (Classical swine fever virus and wild boar migration in South Korea) → Revised title (“Haplotype of Korean wild boars infected by classical swine fever virus subgenotype 2.1d”).

Comment 2: Authors must establish the goal of the manuscript. In the text, the data presented are not sufficient to support the epidemiological considerations, which presuppose a different approach. The result of laboratory examinations is not sufficient for the reported epidemiological considerations. The same applies for the study on the susceptibility of the specific wild boar genotype to CSF, these evaluations require a different experimental design.

Answer: In accordance with the reviewer's comment, we have changed the title such that it better reflects the contents of the manuscript. We have also removed references to CSF transmission caused by wild boar migration.

Comment 3: In the title is mentioned (as in the text) that wild boars are migratory animals, which are not. There are several EFSA opinions on the subject. Please, check.
Answer: As the reviewer points out, the spread of disease is caused not only by the simple movement of wild animals, but also by various transmission factors. The title and manuscript text have been revised accordingly (lines 289–304).

Comment 4: The possible territorial spread of CSFV can be the result of movements of infected pigs / wild boars, products of contaminated pigs or wild boars, people, fomites, vehicles, feed, not only through the movements of wild boars. Given the situation and the spread of CSF in South Korea (the disease is present in domestic pigs and wild boars), the other ways of spreading should also be considered.

Answer: As the reviewer points out, the spread of disease is caused not only by the simple movement of wild animals, but also by various transmission factors. Therefore, the title and content of the manuscript have been revised accordingly (lines 289–304).

Comment 5: Are there structured surveillance plans available whose results allow you to say that the areas where you believe the positivity is shifting were previously disease free?

Answer: Since 2010, about 1500–2000 wild boars have been captured annually nationwide and tested for CSF antigen and antibodies. CSF antigen has not been detected in southern regions; only in the northern regions of South Korea (Gangwon and Gyeonggi provinces). To eradicate CSF in the future, monitoring of wild boars will be performed continuously every year. We have revised the manuscript text accordingly (lines 60–64, 289–295, and Table 1).

Comment 6: It should be explained in what context the samples you refer to are collected.

Answer: To eradicate CSF from the country, the Korean government has been conducting a wild boar monitoring project since 2010, capturing about 1500–2000 animals every year and performing CSF antigen testing. We have revised the manuscript text accordingly (lines 60–64 and Table 1).

Comment 7: References are old and should be updated. As an example, in the European Union countries CSF was eradicated.  Updated data can be found on the WOAH/OIE website.

Answer: We have updated the references and revised the text accordingly (lines 277–286).

Comment 8: Line 45: please, support this statement by a reference

Answer: We have removed the sentence.

Comment 9: Paragraph 3.3: the content does not reflect the name of the title of the paragraph

Answer: We have revised Paragraph 3.3 accordingly (line 187: 3.3. Prediction of CSF occurrence in wild boars).

Comment 10: Line 265-266: did you establish the epidemiological meaning of these positive in relation to the existing wild boar population?

Answer: We have revised these sentences (lines 289–304).

Comment 11: Lines 280-281: This sentence could indicate the presence of the disease in an endemic form in the country.

Answer: This sentence explains that CSF 2.1d, which was prevalent in wild boars, was transmitted to domestic pigs. Before 2016, only CSF 2.1b was present in domestic pigs. In 2016, CSF 2.1d was first detected in domestic pigs. We have revised the text accordingly (lines 311–318).

Comment 12: Lines 284-286: this sentence should be supported by data or specific reference

Answer: We have added references and revised the text accordingly (lines 318–325).

Comment 13: Lines 299-303: the same as above

Answer: We have revised the text accordingly (lines 339–345).

Reviewer 2 Report

The manuscript is relevant, as many countries are currently adopting measures like aggressive hunting to reduce wild boar populations in order to eradicate African swine fever, but these measures can have unintended undesirable consequences. It is generally well written but the English requires some attention from a first language English speaker or professional editor.

Line 45: If CSFV in wild boars in South Korea originated in 2016, what is the explanation for the 2011 isolate? It is actually evident from the literature cited (2 – Kim et al., 2016) that CSFV was known in wild boars in South Korea at least from 2011, and the authors are indicating that the increase in cases in wild boars from 2017 resulted from influx of infected boars from North Korea, but this should be made clearer.

Lines 300 and 310-11: I think they should read that H01 is more susceptible to CSFV – the host is susceptible to the pathogen, not the other way around. I would guess that groups of related wild boars would belong to the same haplotype and that CSFV would be transmitted easily amongst closely associated wild boars, which may be another factor contributing to all the wild boars testing positive belonging to the same haplotype. 

Author Response

Review 2

Comment 1: The manuscript is relevant, as many countries are currently adopting measures like aggressive hunting to reduce wild boar populations in order to eradicate African swine fever, but these measures can have unintended undesirable consequences. It is generally well written but the English requires some attention from a first language English speaker or professional editor.

Answer: This manuscript has been checked by a professional English editing service (www.bioedit.com).

Comment 2: Line 45: If CSFV in wild boars in South Korea originated in 2016, what is the explanation for the 2011 isolate? It is actually evident from the literature cited (2 – Kim et al., 2016) that CSFV was known in wild boars in South Korea at least from 2011, and the authors are indicating that the increase in cases in wild boars from 2017 resulted from influx of infected boars from North Korea, but this should be made clearer.

Answer: This sentence was removed from the revised manuscript.

Comment 3: Lines 300 and 310-11: I think they should read that H01 is more susceptible to CSFV – the host is susceptible to the pathogen, not the other way around. I would guess that groups of related wild boars would belong to the same haplotype and that CSFV would be transmitted easily amongst closely associated wild boars, which may be another factor contributing to all the wild boars testing positive belonging to the same haplotype. 

Answer: We have revised the sentence accordingly (lines 339–345).

Reviewer 3 Report

The manuscript by Choe et al describing CSFV distribution and mtDNA haplotype association is generally well written and clear. However there are a few shortcomings that need to be addressed as certain aspects are not understandable in the current version.

First, 2140 wild boar were screened for CSFV but no prevalence data is given. It says 7 different strains based on the E2 gene were identified but I could not find in the manuscript how many of the boar were positive at all.

This leads to a second “number” issue in that 347 mtDNA Dloop sequences plus 84 reference sequences were obtained from the boar but these individuals are not put into any context with the 2140 samples that were tested for CSFV. If only  a small subset of the CSFV positive samples were among the 347 samples sequenced, then it would be hard to conclude that only the H01 haplotype is CSVF positive since the other positives were not examined. But not knowing how many positives there were or how the 347 samples fit in with this data it is impossible to tell which undermines some of the conclusions of the paper. In the discussion it says the 25 boar with CSFV were all H01 but this does not say if that was the 25 of the 347 that were Dloop typed or if that represents all positives from the 2140.

The authors went to a lot of effort in Figure 2 to describe the spread of CSFV into southern parts of South Korea but then do not discuss the results at all and rather talk about virulence for which no data is presented. The Bayesian skyline plot and figure 2 should be talked about in more detail.

In fact most of the discussion focuses on the situation in Japan and less on explaining the results of the current study and its significance to wild boar and to domestic swine in South Korea. I would recommend re-writing the discussion to discuss the significance of each of the findings in the current study.

Again, I cannot follow the conclusion about mtDNA haplotypes and CSFV as the samples are not matched up between the two data sets and the prevalence of CSFV is not provided.

Author Response

Review 3

The manuscript by Choe et al describing CSFV distribution and mtDNA haplotype association is generally well written and clear. However there are a few shortcomings that need to be addressed as certain aspects are not understandable in the current version.

Comment 1: First, 2140 wild boar were screened for CSFV but no prevalence data is given. It says 7 different strains based on the E2 gene were identified but I could not find in the manuscript how many of the boar were positive at all.

Answer: We have included the prevalence data and a table (lines 60–64, 133–134, and Table 1).

Comment 2: This leads to a second “number” issue in that 347 mtDNA Dloop sequences plus 84 reference sequences were obtained from the boar but these individuals are not put into any context with the 2140 samples that were tested for CSFV. If only a small subset of the CSFV positive samples were among the 347 samples sequenced, then it would be hard to conclude that only the H01 haplotype is CSVF positive since the other positives were not examined. But not knowing how many positives there were or how the 347 samples fit in with this data it is impossible to tell which undermines some of the conclusions of the paper. In the discussion it says the 25 boar with CSFV were all H01 but this does not say if that was the 25 of the 347 that were Dloop typed or if that represents all positives from the 2140.

Answer: Data regarding the total number of pigs tested for CSFV and the number of wild boar subjected to mtDNA tests is presented in Table 1, Figure 3, and in the revised M&M section (lines 91–100, 114–118, Table 1, and Figure 3).

Comment 3: The authors went to a lot of effort in Figure 2 to describe the spread of CSFV into southern parts of South Korea but then do not discuss the results at all and rather talk about virulence for which no data is presented. The Bayesian skyline plot and figure 2 should be talked about in more detail.

Answer: We have revised the text and Figure 2 in accordance with comments made by other reviewers about problems associated with wild boar migration (lines 187–201 and Figure 2).

Comment 4: In fact most of the discussion focuses on the situation in Japan and less on explaining the results of the current study and its significance to wild boar and to domestic swine in South Korea. I would recommend re-writing the discussion to discuss the significance of each of the findings in the current study.

Answer: We have included an explanation in the Discussion section as requested (lines 289-304, 311–325, and 339–345).

Comment 5: Again, I cannot follow the conclusion about mtDNA haplotypes and CSFV as the samples are not matched up between the two data sets and the prevalence of CSFV is not provided.

Answer: Data regarding the number of pigs tested for swine fever and the number of animals in which mtDNA was analyzed are provided as a Table, and the M&M section has been revised to include a better explanation (lines 91–100 and Table 1).

Round 2

Reviewer 3 Report

Thanks to the authors for addressing my comments so thoroughly.

Author Response

Comment 1: Thanks to the authors for addressing my comments so thoroughly.

Answer: Thank you for your review.